# Protective and Therapeutic Capacities of Lactic Acid Bacteria Postmetabolites against Koi Herpesvirus Infection In Vitro

**DOI:** 10.3390/life13030739

**Published:** 2023-03-09

**Authors:** Neli Vilhelmova-Ilieva, Svetla Danova, Zdravka Petrova, Lili Dobreva, Georgi Atanasov, Kapka Mancheva, Lora Simeonova

**Affiliations:** 1The Stephan Angeloff Institute of Microbiology, Bulgarian Academy of Sciences, 26 Georgi Bonchev Str., 1113 Sofia, Bulgaria; 2Institute of Biodiversity and Ecosystem Research, Bulgarian Academy of Sciences, 25 Georgi Bonchev Str., 1113 Sofia, Bulgaria; 3Institute of Biophysics and Biomedical Engineering, Bulgarian Academy of Sciences, 23 Georgi Bonchev Str., 1113 Sofia, Bulgaria

**Keywords:** LAB, probiotics, postmetabolites, Koi herpes virus, antiviral activity, cell protection

## Abstract

Background: The accumulation of data on beneficial biological effects of probiotics and their metabolic products favors their potential use in the prevention and treatment of various malaises. Methods: Nine postmetabolites from Lactic acid bacteria (LAB) of human or dairy origin and their antiviral activity were studied using the cytopathic effect inhibition test. The virucidal capacity, their influence on the adsorption stage of Koi herpes virus (KHV) and their preventive role against subsequent viral challenge on intact Common carp brain (CCB) cells were also determined by titration assay. Residual viral infectivity in postmetabolites-treated samples was compared to mock-treated controls and Δlgs were calculated. Results: When administered during KHV replication, the microbial products isolated from *Lactiplantibacillus plantarum* showed remarkable activity with a selectivity index (SI) between 26.5 and 221.4, as those effects were dependent on the sample-virus incubation time. Postmetabolites from *Lactobacillus gasseri* and *Lactiplantibacillus plantarum* also demonstrated significant inhibition of KHV replication with SI of 24 and 16, respectively. The bioactive metabolites isolated from *Limosilactobacillus fermentum* had a minor effect on the viral replicative cycle. Compounds, produced during the fermentation by lactobacilli, grown on different nutritive media and collected at different time points, significantly inhibited extracellular KHV virions. All investigated postmetabolites remarkably blocked KHV attachment to the host cell (CCB), leading to a drop in viral titers by Δlg = 4.25–5.25, and exerted protective effects on CCB cells before they were subjected to viral infection. Conclusions: Our results open new horizons and promote LAB and their postbiotic products to be used in the prophylaxis and therapy of viral infections.

## 1. Introduction

Probiotics are live microorganisms, having particular properties and life cycle, that contribute a number of beneficial effects to the microorganism. When applied in adequate amounts in appropriate conditions, these prokaryotes, species- and strain-dependently, produce various substances, called postbiotics, which improve the host’s biological functions. The International Scientific Association for Probiotics and Prebiotics (ISAPP) defined a postbiotic as “a preparation of inanimate microorganisms and/or their components that confers a health benefit on the host” [1,2].

By date, scientifically accredited and commercially available probiotics belong to the genus *Bifidobacterium* and family *Lactobacillaceae* with species recently re-classified into 23 new genera [3]. On the other hand, prebiotics are specialized plant fibers found in fruits and vegetables that stimulate the growth and activity of probiotics. By combining prebiotics and probiotics, synbiotics are obtained, which when applied in this way have exceptional health benefits. Lactic acid bacteria (LAB) are generally considered safe due to their long-term utilization in food fermentation systems and as potent producers of bioactive compounds (hydroxyl fatty acids, phenyllactic acid, hydrogen peroxide, bacteriocins, and carbon dioxide) with a wide spectrum of antimicrobial action [4,5,6].

Currently, prebiotics and probiotics are the predominantly marketed products, but postbiotics become a subject of increasing attention and scientific research. They comprise a wide range of non-viable biochemical molecules released by probiotics through a fermentation process and/or produced in pure form on a laboratory scale (such as teichoic acids, peptides, enzymes, peptidoglycans, polysaccharides, organic acids and extracellular proteins) [7,8]. These substances have natural biological activities that have been extensively studied in recent years. Immunomodulatory, anti-inflammatory, anticarcinogenic, antibacterial, antiviral, anticancer, antioxidant, antidiabetic, obesity-reducing, antihyperhypertensive and antiallergic efficacies constitute the most important biological roles of postbiotics [9,10]. In terms of safety, it has been confirmed that postbiotics, as potential substitutes, may be even superior to their parental living cells. In addition, due to their suitable economic, technological and clinical characteristics, they can be used as beneficial raw materials in the food and pharmaceutical industries as supplements [9,11].

Staying focused on human benefit and care, in the concept of One Health, the investigation of important animal species with a significant social and economic impact, such as various aquacultures, appears to be essential. Koi Cyprinus carpio herpesvirus infection, also known as viral nephritis and gills necrosis, is a detrimental disease in carp and carp’s ornamental species caused by koi herpesvirus (KHV). The latter is an enveloped DNA virus, also known as Third Cyprinid Herpesvirus (CyHV-3), which is highly contagious and can cause up to 100% mortality in carp population, leading to huge economic losses in fish industry. Koi herpesvirus disease (KHVD) was first reported in Israel in May 1998 and is currently distributed worldwide with the exception of Australia [12,13]. By affecting both wild and cultured populations of common and koi carp it is one of the most commonly spread infections in fish and it is on the European Union (EU) list of exotic diseases under surveillance by the European Reference Laboratory for Fish Diseases. [14]. Despite scientific efforts, prophylaxis and a treatment scheme of KHVD have not been approved so far [15]. Factors like immune system evasion by the virus, the pressure of antivirals leading to selection of resistant variants, drug toxicity and side effects, their economic cost, delivery and disposal necessitate finding novel treatment approaches which are effective but also host- and environment-friendly.

Specific health-promoting effects of microbial products in treatment of both infectious and non-infectious diseases have been described in the literature. If applied in cases of diarrhea, they alleviate gastrointestinal discomfort, nutrients’ absorption and assist B and K vitamins production, as well as reducing the risk of viral infections in humans [16,17,18,19,20,21,22]. The role in infections has been studied in vitro and in vivo, showing highly promising results of probiotic and their postmetabolites treatments against a number of bacterial and viral pathogens such as Salmonella, Influenza, HSV type 1, Rota-, Noro-, ECHO viruses [23,24].

Huge work has been done in the research and results being published on probiotic microbial strains effects on human health, but data on the potential to bring an advantage to animal well-being, including prevention and treatment of diseases, remains limited.

Although fish viral diseases are difficult to prevent and treat, there are some observations on the profits of probiotic supplementation. Published data reveal that orally administered single or a mixed probiotic strains could improve the health condition of common carp. Different *Lactobacillus plantarum* strains showed potential, as fish probiotics increase growth rate and enhance the immunity of common carp (*C. carpio*) and koi carp (*C. rubrofuscus*) against koi herpesvirus, but additional studies are needed [25]. Recently, Huang et al. showed that chitosan–alginate capsules as an oral delivery system for a live probiotic (*Lactobacillus rhamnosus*) vaccine, expressing KHV ORF81 protein, protected 85% of koi carps infected with KHV [26].

Despite the positives of probiotics, it has been proven that as a result of their viability, these bacteria can induce undesired effects under certain circumstances. Supplements containing live probiotic bacteria do not necessarily have the same beneficial effects in subjects of different ages and physical conditions. Quite often, these microbes lead to adverse reactions, especially on a background of weakened immune function, thus causing clinical complications in patients with Crohn’s disease, pregnant women, the elderly and infants [27]. Another drawback to the use of probiotics is the emergence of antibiotic resistance and the potential for transmission of genes that cause virulence and resistance from pathogens found in the host’s gut. There are also opportunistic pathogenic bacteria in the gut microflora and the acquisition of antibiotic resistance can be associated with serious problems [28]. Various strategies have been tried in recent years to overcome these adverse events. One of the most effective and practical approaches could be the use of metabolites/inactivated form of bacteria (postbiotics) as substitutes of living probiotics [9,29].

Herein, we investigated the activity of nine postmetabolites of viable lactic acid bacteria isolated from various dairy products on koi herpes virus replication steps in vitro.

## 2. Materials and Methods

### 2.1. Lactobacilli and Postmetabolites Tested

Four *Lactobacillus strains* were pre-selected from the laboratory collection of the Institute of Microbiology as follows: (i) with human origin—*Limosilactobacillus fermentum 54S, Lactobacillus gasseri VS, Lactiplantibacillus plantarum* HS; and (ii) *Lactiplantibacillus plantarum* L3 from fermented milk product katak. *Lactobacillus* cultures were stored in MRS broth (Merck, Darmstadt, Germany) supplemented with 20% *v/v* glycerol at −20 °C. Prior to the experiments, they were passaged twice in MRS broth (Merck, Darmstadt, Germany) at 37 °C, followed by fermentation in MRS broth or in HiVeg MRS broth (HiMedia, Mumbai, India), only the strain L3, and in skimmed milk (reconstituted 10% *w/v* Fluca) (as shown in Table 1).

The active postmetabolites were obtained from *Lactobacillus* cultures at 24 h, 48 h and 96 h. The cells were harvested by centrifugation (5 min, 5500 rpm, HermLe Centrifuge, Wehingen, Germany) and the cell-free supernatants (CFS) were collected and stored at −20 °C. Prior to the experiments, they were filtered using syringe filters Millipore (0.22 µm). The CFS from exponential (24 h) and stationary phase (48 h) cultures of *L. plantarum* L3, at 37 °C in MRS broth, were collected as well.

The whey fraction was obtained from sterile skimmed milk (Serva, 10% *w/v*) inoculated with 5% inoculum of an exponential culture of L3 strain. Fermented overnight milks were centrifuged (3000 rpm, HermLe Centrifuge, Wehingen, Germany) for 10 min, and supernatant was collected and stored at −20 °C. Two-stage fermentation was also applied as followed: 1st stage overnight cultivation of L3 skimmed milk (Humana, Bremen, Germany) and obtained fermented milk was used as 5% (*v/v*) inoculum in MRS broth (HiMedia, Mumbai, India) for 12 h at 37 °C. Obtained *L. plantarum* L3 exponential culture MRS was inoculated (10% *v*/*v*) in fresh MRS broth to obtain 48 h CFS in MRS (HiMedia, Mumbai, India) medium (Sample L3C, Table 1).

### 2.2. Virus and Cells

Common carp brain (CCB) cell line was provided by the European Union Reference Laboratory for Fish and Crustacean Diseases, National Institute of Aquatic Resources Technical University of Denmark. They were cultivated in growth Eagle’s Minimal Essential Medium (EMEM) (Gibco BRL, Paisley, Scotland, UK), which contained 2 mM Glutamine, 1% Non Essential Amino Acids (NEAA), 10% Fetal Calf Serum (Gibco BRL, Carlsbad, CA, USA), 10 mM HEPES buffer (Merck, Germany) and antibiotics (penicillin 100 IU/mL, streptomycin 100 μg/mL) in a CO_2_ incubator (HERA cell 150, Heraeus, Hanau, Germany) at 22 °C/5% CO_2_.

The koi herpes virus (KHV), F-347 strain was purchased from the American Type Culture Collection (ATCC) and propagated in CCB cells in a maintenance eagle’s medium, with 2% fetal bovine serum added at 22 °C and 5% CO_2_. The viral infectious titer was quantified as 10^6.5^CCID_50_/mL.

### 2.3. Cytotoxicity Assay

Confluent CCB monolayers were overlayed with 0.1 mL/well EMEM containing no (untreated control)/or descending concentrations of postmetabolites in 96-well plates (Costar^®^, Corning Inc., Kennebunk, ME, USA) and were housed at 22 °C and 5% CO_2_ for 3 days. On the 72nd hour, the medium with the test samples was discarded, cells were washed with buffered saline solution (PBS), dyed with neutral red (NR) and incubated at 37 °C for 3 h. The NR was aspirated, with cell monolayers washed again with PBS, and subjected to a desorbing aqueous solution of 1% glacial acetic acid and 49% ethanol. Each well was exposed to 540 nm and optical density (OD) was measured in a microplate reader (Biotek Organon, West Chester, PA, USA). The concentration that destroyed cells by 50% when compared to mock-treated control represents the sample’s CC_50_ (50% cytotoxic concentration). The test was performed in triplicate with four wells per sample.

### 2.4. Antiviral Activity Assessment

Postmetabolites’ antiviral activity was assessed by their capacity to inhibit cytopathic effect (CPE) arising from KHV multiplication in confluent cells infected with 0.1 mL viral suspension at a multiplicity of 100 CCID_50_ (cell culture infectious dose 50%). Following 1 h of virus adsorption, the virus was removed, and cells were treated with serial dilutions of each substance, being then incubated at 22 °C for 72 h. The extent of cell damages, induced by the virus was evaluated using a neutral red uptake assay, as the reduction of CPE for each concentration of the test sample was calculated in percent by the following formula:% CPE = [ODtest sample − ODvirus control]/[ODtoxicity control − ODvirus control] × 100,
where OD_test sample_—means OD value of the wells treated with the virus and test sample in the respective concentration, OD_virus control_—means OD value of the wells with no postbiotic treatment (virus control) OD_toxicity control_—means OD value of the wells with no virus but treated with postmetabolite at the corresponding dilution.

The concentration of the sample that inhibited 50% CPE when compared to the virus control was defined as IC_50_ (50% inhibitory concentration). The ratio CC_50_/IC_50_ represents SI (selectivity index), which is the major indicator for antiviral capacity of each tested product in the experiment.

### 2.5. Virucidal Effect

KHV (10^4^ CCID_50_) was combined with each of the tested compounds at its MTC in a 1:1 ratio in samples of 1 mL, which were kept at room temperature for 15, 30, 60 and 90 min. After each contact interval, the residual infectious content in the sample was titrated in CCB cell cultures and Δlgs were determined in relation to the untreated controls.

### 2.6. Virus Adsorption Test

Common carp brain cells grown in twenty four-well plastic ware were pre-chilled at 4 °C and subsequently infected with 10^4^ CCID_50_ of KHV concomitantly being subjected to MTC of the tested postmetabolites for 1 h at 4 °C. At various time points (15, 30, 45 and 60 min), cell monolayers were washed with PBS to discard both the sample and the unadsorbed virus. They were overlaid with maintenance EMEM and incubated at 22 °C for 24 h. After consecutive triple freezing and thawing to destruct cellular integrity, the infectious titer of each sample was examined by the end-point dilution method.

### 2.7. Pre-Treatment of CCB Cells

CCB cells seeded in 24-well culture plates (CELLSTAR, Greiner Bio-One, Kremsmünster, Austria) (2 × 10^6^ cells per well) were treated for 15, 30, 60, 90 and 120 min at MTC of the postmetabolites in a maintenance medium (1 mL/well). The cells were washed with PBS and challenged with 1000 CCID_50_/mL per of KHV. Following 60 min of virus attachment, the non-adsorbed virus was removed and the cells were covered with the maintenance medium. The plates were incubated at 37 °C for 24 h and, after triple freezing and thawing, the infectious viral titers were determined by the end-point dilution method.

### 2.8. Statistical Analyses

All antiviral tests were performed as three independent experiments under the specified conditions. CC_50_ and IC_50_ values were presented as means ± standard deviations (SD). The comparison of postmetabolites’ toxicity and their effects on KHV replication to those of the reference nucleoside analogue acyclovir (ACV) was performed by Student’s *p*-test, and *p*-values of <0.05 were considered significant. The accumulated raw datasets on samples’ cytotoxicity and antiviral effects were analyzed statistically by Graph Pad Prism 4^®^ software.

## 3. Results

### 3.1. Cytotoxicity

We were interested in testing if postbiotics could be more efficient or possess different antiviral effects by changing the conditions of cultivation (such as media) and the environment, as well as the fermentation time. For this purpose, we collected samples S1 to S5 produced by *L. plantarum* L3 from Katak and characterized in our previous study with high activity against HSV-1 and now cultivated in various media and conditions. Commercial medium MRS and a vegan version of the medium MRS-Hi veg (Table 1) were used. The testing comprised samples obtained on the 24th, 48th and 96th hours of the fermentation process. All studied LAB postmetabolites showed weak and similar cytotoxicity against cells of the CCB line, significantly lower than that of ACV (Table 2).

### 3.2. Antiviral Activity

The most significant influence on KHV replication was shown by the group of L3 samples. The activity of S2 with SI = 221.4 is the clearest, and the S1 with SI = 214.3 has similar activity. Postmetabolites S3 (SI = 180.4) and S4 (SI = 164) were slightly less active but still significantly inhibit viral replication. S5 also demonstrated antiviral activity, but several times weaker than the other representatives of the L3 group. Similar activity to S5 was demonstrated by S6 with SI = 24.0. Some antiviral activity was also exhibited by S9 samples (SI = 16.0), while inhibition by S7 and S8 was negligible (Table 2).

### 3.3. Virucidal Effect

In addition to the influence on the KHV replicative cycle, the influence of postmetabolites on extracellular KHV virions was reported. The inhibitory effect was monitored at four time intervals: 15, 30, 60 and 90 min. No data were obtained from longer intervals because KHV virions completely lost their infectivity on their own after 120 min at room temperature in the absence of cells. At 15 min of exposure, none of the samples had significant virucidal activity. At 30 min, only S3 showed distinct activity and weak inhibition was demonstrated by S1, S2 and S6. A tendency is observed—with increasing exposure time, the activity of the investigated postmetabolites also increases. At 90 min of exposure, a significant decrease in viral titers with Δlg = 2.0 was shown by samples S1, S2, S3, S6 and S9. Sample S4 also exerted significant virucidal activity (Δlg = 1.75). The effect of S7 and S8 was insignificant. MRS broth was used as a negative control, and 70% ethyl alcohol as a positive control (Table 3).

### 3.4. Influence on the Stage of Viral Adsorption

The effect of postmetabolites on the attachment of virions to susceptible CCB cells is summarized in Table 4. The most distinct inhibition of the process was reported by S1, S2, S3, S4, S6, S7 and S9 at 15 min of exposure, and the virus titer was reduced by Δlg = 2.0. In the next time point, i.e., 30 min, all investigated compounds prevented viral adsorption to a varying extent, with the most pronounced effect at S2 and S6 with a decrease in viral titer by Δlg = 3.0. The adhesion becomes more strongly suppressed with the exposure time for all the tested samples. The most significant effect was demonstrated by S2 (Δlg = 5.25) followed by S1, S3, S4, S6, S7 and S8 (Δlg = 5.0) at 60 min of contact (Table 4).

### 3.5. Pre-Treatment of CCB Cells

The high efficacy that the postmetabolites showed on the adsorption stage of the virus to sensitive cells if compared to the virucidal one suggests that the studied samples have an impact on structures included in the composition of the cell membrane. To follow up this interaction, we examined the influence of postmetabolites when they were administered before virus infection on CCB cells in the absence of pathogen. After 15 min of incubation of LAB samples with uninfected cells S2 (Δlg = 2.5), S3 and S4 (Δlg = 2.0) and to a lesser extent S1, S5, S6, S7 and S8 (Δlg = 1.75) significantly protected cells from subsequent KHV infection. The protective effect on healthy cells increases with exposure time. At the last monitored time interval of 120 min, all postmetabolites demonstrated a significant effect, the most noticeable being in the S4 (Table 5).

## 4. Discussion

Acquired resistance to antiviral agents as a result of their application is a growing problem worldwide, as respiratory, sexually transmitted, enteric viruses have been reported in humans, as well as a large number of viruses causing economic losses due to severe crop damage or on farm animals and in fish farms. Therefore, there is a constant search for new, non-traditional antiviral agents that can serve as an alternative to currently used drugs. In recent years, data on the efficacy of LAB and their bacteriocinogenic postmetabolites as antiviral agents have been increasing. Health-beneficial LAB probiotics may exert their antiviral activity through (1) direct probiotic–virus interaction; (2) production of antiviral inhibitory metabolites; and/or (3) stimulation of the immune system [30].

Probiotics and their metabolic products might disturb viral infections indirectly by stimulating the adaptive and innate immunity. The activated immune response can reduce the duration and severity of the disease. It also improves the generation of virus-specific antibodies and normalizes intestinal permeability [31]. The antiviral effect of postbiotics depends on the type of probiotic used to extract them, as well as the type of virus. It has been proven in vitro that postbiotics exert antiviral effects when they meet with enveloped viruses. They can prevent viral absorption and penetration into the host cell, which are initial but crucial steps of the infection. The interference with virus entry into the host cell occurs likely by blocking viral binding to host cell receptors [32]. A study demonstrated the effect of cell-free metabolites of *Lacticaseibacillus rhamnosus* on enterovirus and Coxsackievirus in HeLa, Vero and Hep-2 cell lines. *L. rhamnosus* metabolites were found to interrupt virus attachment to cell lines [33]. With proven inhibition of HSV-2 entry into cells, researchers have hypothesized that *Lactobacillus crispatus* forms a protective biofilm on the cell surface that blocks HSV-2 receptors and thereby prevents the virus from entering cells [34]. The antiviral mechanism of organic acids produced by probiotic bacteria has been observed through their binding to the glycoproteins of viruses, thus again disrupting viral entry [35]. In addition, some authors suggest that probiotics can also affect the later stages of the viral cycle in cells [36].

The results described by other teams are consistent with our results [32,33,34,35,37]. A large proportion of the investigated postmetabolites demonstrated an effect on the enveloped extracellular Koi herpesvirus particles. All of the products we tested showed remarkable inhibition of the viral adsorption step to CCB cells, with reductions in viral titers above Δlg > 4–5.

Pretreatment of cells with metabolic products of the probiotic *L. plantarum* strain N4 against transmissible gastroenteritis coronavirus showed a significant protective effect on the cells [38]. This study supports the results obtained by us upon pretreatment of healthy CCB cells with the studied postmetabolites, which revealed a significant decrease in viral titers upon the subsequent koi herpesvirus infection. The protective effect of the tested samples on uninfected cells can be explained by the bacteriocins contained, which can block the host cell receptors responsible for the recognition and binding of the virus [39]. Consistent with current data on experimental KHV our previous study proved the antiherpetic activity of postmetabolites isolated from different LAB strains against HSV-1. Some of the products had significant effects on viral replication, but more notable was their effect on the adsorption stage, as well as on extracellular virions. Similarly, a protective effect on MDBK cells subsequently subjected to HSV-1 infection was reported [24]. Our results once again confirm previously reported data on the anti-herpes simplex virus type-1 (HSV-1) activity of LAB-postmetabolites. Although the two studied viruses HSV-1 and KHV belong to different families and have certain differences in the stages of their replication, the external viral architecture is alike, and similarly, the postmetabolites exert their influence outside the cell, mostly acting on the stage of viral adsorption.

## 5. Conclusions

Based on our study results, it can be concluded that the postmetabolites isolated from *L. plantarum* and *L. gasseri* significantly affect the replication of KHV, influence extracellular virions and exert a remarkable effect on the stage of virus adsorption to CCB cells.

Findings on LAB and derived postbiotics have capacities to reduce damages caused by both human and koi herpesvirus pathogens, highlighting these compounds as powerful candidates for inclusion in prophylaxis and treatment in human and veterinary medical practice, but further data are needed for the assessment of their antiviral potentials in vivo.

## Figures and Tables

**Table 1 life-13-00739-t001:** Lactobacillus samples tested.

Designation	Sample	Specifications
S1	L3-24 h MRS	Filtered cell-free supernatants (CFS) from exponential culture (24 h) of strain *Lactiplantibacillus plantarum* L3 in MRS broth (Merck, Germany)
S2	L3-48 h MRS	CFS from stationary phase culture (48 h) of strain *L. plantarum* L3 in MRS broth (Merck, Germany)
S3	L3-48 h HiVegMRS	CFS from stationary phase culture (48 h) of strain *L. plantarum* L3 in MRS broth (HiVeg, HiMedia, India)
S4	L3- C	CFS from stationary phase culture (48 h) of strain *L. plantarum* L3 in MRS broth (HiMedia, India)
S5	L3-96 h	CFS first bouillon in MRS broth (HiVeg, HiMedia, India) from late stationary phase culture (96 h) of the strain *L. plantarum* L3
S6	*L. gasseri* VS	Filtered CFS from exponential culture (24 h) of strain *Lactobacillus gasseri* in MRS broth (Merck, Germany)
S7	Lf54S-MRS	Filtered Cell-free supernatants (CFS) from exponential culture (24 h) of strain L54S in MRS broth (Merck, Germany)
S8	Lf54S-WF	Whey fraction from fermented sterile skimmed milk (10% *w/v* Fluka) with strain *Limosilactobacillus fermentum* Lf54S
S9	Lpl. HS	Filtered Cell-free supernatants (CFS) from exponential culture (24 h) of strain *Lactiplantibacillus plantarum* 2HS in MRS broth (Merck, Germany)

**Table 2 life-13-00739-t002:** LAB postmetabolites’ cytotoxicity and antiviral activity in vitro.

Tested Samples	Cytotoxicity (µg/mL)		Antiviral Activity	
	CC_50_	MTC	IC_50_ (µg/mL)	SI
S1	7.5 ± 0.7 ***	1.0	0.035 ± 0.002 ***	214.3
S2	6.2 ± 0.9 ***	1.0	0.03 ± 0.006 ***	221.4
S3	8.3 ± 0.8 ***	5.0	0.05 ± 0.003 ***	180.4
S4	8.2 ± 0.7 ***	5.0	0.05 ± 0.002 ***	164.0
S5	5.3 ± 0.4 ***	1.0	0.2 ± 0.009 ***	26.5
S6	4.5 ± 0.2 ***	1.0	0.18 ± 0.008 ***	24.0
S7	3.1 ± 0.2 ***	1.0	1.72 ± 0.04 ***	1.8
S8	6.5 ± 0.6 ***	1.0	1.22 ± 0.02 ***	5.3
S9	6.4 ± 0.5 ***	1.0	0.04 ± 0.002 ***	16.0
MRS broth	-	-	-	-
ACV	820.0 ± 6.8	320.0	16.2	50.6

Student’s *p*-test: *** *p* ˂ 0.0001 as compared to reference ACV.

**Table 3 life-13-00739-t003:** Virucidal activity against KHV various of LAB postmetabolites/parabiotics.

Tested Samples			Δlg	
	15 min	30 min	60 min	90 min
S1	1.0	1.5	2.0	2.0
S2	1.0	1.5	2.0	2.0
S3	1.25	1.75	2.0	2.0
S4	1.0	1.0	1.5	1.75
S5	1.0	1.0	1.5	1.5
S6	0.75	1.25	1.75	2.0
S7	0.25	0.25	0.5	0.75
S8	0.25	0.5	0.75	1.0
S9	1.0	1.0	1.5	2.0
MRS broth	0	0	0	0
70% ethyl alcohol	6.0	5.5	4.75	4.25

**Table 4 life-13-00739-t004:** Effect of LAB postmetabolites on viral adsorption of KHV on CCB cells.

Tested Samples			Δlg	
	15 min	30 min	45 min	60 min
S1	2.0	2.75	4.25	5.0
S2	2.0	3.0	4.5	5.25
S3	2.0	2.5	4.5	5.0
S4	2.0	2.5	4.5	5.0
S5	1.5	2.5	3.75	4.75
S6	2.0	3.0	4.0	5.0
S7	2.0	2.5	4.0	5.0
S8	2.0	2.75	4.0	5.0
S9	2.0	2.75	4.0	4.25
MRS broth	0	0	0	0

**Table 5 life-13-00739-t005:** Pretreatment of CCB cells with LAB-derived fragments or postmetabolites before KHV infection.

Tested Samples			Δlg		
	15 min	30 min	60 min	90 min	120 min
S1	1.75	1.75	2.0	2.0	2.0
S2	2.5	2.5	2.5	2.5	2.5
S3	2.0	2.0	2.0	2.0	2.0
S4	2.0	2.5	2.5	3.0	3.0
S5	1.75	1.75	1.75	1.75	1.75
S6	1.75	2.0	2.0	2.5	2.5
S7	1.75	1.75	2.0	2.0	2.0
S8	1.75	1.75	2.0	2.0	2.0
S9	1.25	1.25	1.5	1.75	2.0
MRS broth	0	0	0	0	0

## Data Availability

Not applicable.

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
