# Peer review of "Protective and Therapeutic Capacities of Lactic Acid Bacteria Postmetabolites against Koi Herpesvirus Infection In Vitro"

_life, 2023, doi:10.3390/life13030739_

Round 1

Reviewer 1 Report

1. In title, the authors used the words LAB (Lactic acid bacteria) which was not commonly used and difficult to understand for the general audience. It should be long form. Please revise it.

2. At page-2, second line, the reference no-2 font size is larger than the other font size. Please revise it. Why did the authors showed the red colour in font for some word? Is it any specific aims to showed like this? 

3. Regarding the study on Herpes Virus, why did the authors choose Herpes virus groups in staed of other viruses. Although the authors wrote a long introduction for this manuscript, the reviewer could not find any justification for choosing on Herpes virus group. Please make justification for choosing this viruses and add at the revised manuscript.

4. Regarding the Koi Herpes virus , please add the Genbank accession no.

5. Regarding the postmetabolites, how did the authors identify the active ingredients were including at your postmetabolites constituents? I did not see any quality assurance procedures for the postmetabolites constituents in this study. Please add and discuss at the revised manuscript.

6. Herpes viru group has antiviral drug such as acyclovir. The reviewer did not see any positive control when the authors did experiments about the virucidal activity of post metabolites of LAB. Please describe and justification for not adding the positive control.

7. Regarding the virucidal activity of a compound, the researches need the confirmation of the supernatant of cell culture fluid by means of virus titration assay. The reviewer only see the virucidal acitivity investigation by means of reduction of CPE activity. Is it possible to make conclusion?  

Reviewer 2 Report

This paper described the antiviral function of postbiotics in carp cells. It could be interesting in the veterinary field and could serve as a guiding opinion in fishery medicine. Here are my comments and concerns:

1. The abstract section is too disorganized, and it's not clear which parts are background information and which parts are the author's results. Please organize it according to the structure of background, methods, results, and conclusion.

2. The introduction section is too general. The author used most of the space to introduce how important postbiotics are, but only used limited space to explain the role of postbiotics in their own research topic. Suggestions: (1) Provide more information about the severity of KHV in the fishery, and (2) Discuss existing research on postbiotics and their antiviral effects.

3. Section 2.1 and Table 1 employed various MRS media (such as S2: MRS from Merck, S3: MRS from HiMedia and HiVeg brand, S4: MRS from HiMedia without a brand). Could you clarify the distinctions among them and the reason for conducting the comparison?

4. Section 2.3, the last sentence is confusing. Please rephrase.

5. Section 2.7, please explain what is ∆lg.

6. Please split the results section into different subsections based on the aspects (antiviral activity, virucidal activity, virus absorption, etc.), otherwise it will be difficult to follow.

7. Table 2, what is ACV?

8. Table 3, though the authors did not explain what is ∆lg, it looks like it is an index of virucidal activity. The higher the number is, the virucidal activity is stronger, while 1 means no virucidal activity. If this is true, Please explain why the S7 and S8 have shown an enhanced effect on virus infection.

9. Table 3~5, control is missing for this experiment. The best control in my mind is the postmetabolites of an irrelevant but non-toxic bacteria. But at least, please use the fresh bacteria medium as control.

10. The Conclusion section, (1) please place it in front of Discussion, and (2) please describe more about your conclusion. For example, what is the best strain and condition combination to generate the most effective virucidal postbiotics?

Author Response

Dear Reviewer,

Thank you for your questions, comments and remarks. We find them as completely reasonable and we believe that they will contribute to improving the quality of our manuscript.

Our answers with the corresponding questions are:

  1. The abstract section is too disorganized, and it's not clear which parts are background information and which parts are the author's results. Please organize it according to the structure of background, methods, results, and conclusion.

Answer: The abstract is organized according to the structure of background, methods, results, and conclusion.

  1. The introduction section is too general. The author used most of the space to introduce how important postbiotics are, but only used limited space to explain the role of postbiotics in their own research topic. Suggestions: (1) Provide more information about the severity of KHV in the fishery, and (2) Discuss existing research on postbiotics and their antiviral effects.

Answer: We have presented in the introduction more information about the importance of KHV on a global scale. We have also added some recent studies on the use of probiotics in the prevention and antiviral therapy of KHV diseases.

  1. Section 2.1 and Table 1 employed various MRS media (such as S2: MRS from Merck, S3: MRS from HiMedia and HiVeg brand, S4: MRS from HiMedia without a brand). Could you clarify the distinctions among them and the reason for conducting the comparison?

Answer: Lactobacilli may produce different antimicrobials. Our previously production studies showed that metabolic activity, respectively spectrum of active compounds, during the fermentation depend from culture’s conditions – time, temperature and compound of nutrition media. Therefore, we decide to compare two brands of selective for LAB - De Man Rogosa Sharp broth medium – MRS (Merck, Germany and HiMedia, India both with peptone from casein) with MRS (HiVeg, India, wich is a vegan version of the MRS). Moreover, the MR (HiMedia -India) brand differed in component (type of peptone, which is important for the LAB growth) and is recommended as medium for Lactiplantibacillus plantarum.

The sample 4 (S4) - missing brand was added. Thank you.

  1. Section 2.3, the last sentence is confusing. Please rephrase.

Answer: The sentence is rephrase.

  1. Section 2.7, please explain what is ∆lg.

Answer: An explanation of what Δlgs means is given in the text.

  1. Please split the results section into different subsections based on the aspects (antiviral activity, virucidal activity, virus absorption, etc.), otherwise it will be difficult to follow.

Answer: The results are separated into subsections.

  1. Table 2, what is ACV?

Answer: ACV is an acyclovir nucleoside analogue used in the treatment of herpes infections. The explanation is supplemented in materials and methods in the subsection "Statistical analysis" - at the beginning of page 7.

  1. Table 3, though the authors did not explain what is ∆lg, it looks like it is an index of virucidal activity. The higher the number is, the virucidal activity is stronger, while 1 means no virucidal activity. If this is true, Please explain why the S7 and S8 have shown an enhanced effect on virus infection.

Answer: S7 and S8 did not show an enhanced effect on virus infection. Their activity is insignificant and this is indicated at the end of the paragraph on page 8.

  1. Table 3~5, control is missing for this experiment. The best control in my mind is the postmetabolites of an irrelevant but non-toxic bacteria. But at least, please use the fresh bacteria medium as control.

Answer: MRS medium was used as a control in all experiments. We did not consider it necessary in the absence of effect on the virus to report the result in determining the change in infectious viral titers Δlgs. We have taken note of your remark and included the result in Tables 3 – 5. When determining the virucidal activity (Table 3) was used as a positive control and 70% ethyl alcohol and is supplemented in the table.

  1. The Conclusion section, (1) please place it in front of Discussion, and (2) please describe more about your conclusion. For example, what is the best strain and condition combination to generate the most effective virucidal postbiotics?

Answer: Part of the conclusion has been moved to the discussion, and the conclusion itself has been rewritten to present more specifically the most active of the studied postmetabolites.

We hope we've answered your questions and fulfulled your recommendtions.

Reviewer 3 Report

The manuscript „Protective and therapeutic capacities of lab postmetabolites against Koi Herpesvirus infection in vitro“ by Vilhelmova-Ilieva et al., validates different postbiotics produced by L. plantarum by defining their effects on KHV replication and viral adsorption.

I would like to thank the authors for providing this interesting manuscript. Please find major and minor points that the authors should address.

Major points:

I do not understand the procedure of the antiviral activity assessment. Why was an MOI of 100 chosen? Wouldn´t it make more sense to only use a low MOI, especially when a total incubation time of 72 h was used? Furthermore, why did the authors only asses the IC50 via the visible CPE. Wouldn´t it be more accurate to perform a plaque assay? I would appreciate if the authors would calculate the IC50 values by plaque assay.

The data of the CC50 and IC50 determination should be shown in more detail as an supplementary file. It is hard for the reader if only the final values are given in tables. Why haven´t the authors prepared some graphs to show the actual data?

It would be good to also evaluate the simple MRS media for a putative antiviral effect. Maybe it was done while evaluating the IC50, but due to missing data one cannot tell.

Concerning the evaluation of the virucidal activity shown in Table 3, is there any possibility to use a positive control. It would make the data more reliable than only showing the postmetabolites virucidal activity.

Furthermore, as the virions seem to be quite unstable the authors should provide data to show that after 90 minutes the viruses are still fully infective when not incubated with the substances. It needs to be clarified that the substances to have a virucidal activity at time points that do not change the infectivity per se.

Why was the pre-treatment of CCB cells performed at 37°C while all other experiments were done at 22°C.?

Minor comments:

Page 2: “They comprise a wide range of non-viable biochemical molecules relased by…”

Released instead of relased

Page 2: “…antihyperhypertensive…” change to antihypertensive

Page 5: “Cytotoxicity assay:  On 72th hour…” change to 72nd

Page 11: “Discussion: …glycoproteins of viruses thus again disrupting vual entry.” Change vual in viral.

Page 11: “Discussion: The results described by other teams are consistent with our results” The authors need to provide some literature.

Author Response

Dear Reviewer,

Thank you for your questions, comments and remarks. We find them as completely reasonable and we believe that they will contribute to improving the quality of our manuscript.

Our answers with the corresponding questions are:

Major points:

I do not understand the procedure of the antiviral activity assessment. Why was an MOI of 100 chosen? Wouldn´t it make more sense to only use a low MOI, especially when a total incubation time of 72 h was used? Furthermore, why did the authors only asses the IC50 via the visible CPE. Wouldn´t it be more accurate to perform a plaque assay? I would appreciate if the authors would calculate the IC50 values by plaque assay.

Answer: In the experiment, MOI = 100 was not chosen, but 100 CCID50 (cell culture infectious dose 50%), which infects the cells in the entire well. This makes the multiplicity of infection well below 1 (MOI = 0.005).

The IC50 was determined only by determining the cytopathic effect exhibited because the Koi herpesvirus F-347 strain did not make clearly distinguishable plaques on the CCB cell monolayer. Viral microplaques are present in places, but in general the virus invades the monolayer simultaneously and the individual plaques appear fused and cannot be assessed. So it would be extremely difficult or almost impossible to determine the IC50 by the plaque assay.

The data of the CC50 and IC50 determination should be shown in more detail as an supplementary file. It is hard for the reader if only the final values are given in tables. Why haven´t the authors prepared some graphs to show the actual data?

Answer: When determining a selective substance index (SI) it is better to present the CC50 and IC50 values from which the SI value is derived. Graphical representation is more often used when examining one or two substances. The aim of our research is 9 postmetabolites, and if the corresponding cellular or viral control is also presented in the figure, there should be 10 intertwining curves, from which the reader has no way to orient himself for the CC50 and IC 50 values. That is why we have presented the results in a table with the ready values. I present research from other scientific teams who present the results as we do.

Zhihui Li, Bin Cui, Xiaowen Liu, Laicheng Wang, Qingjie Xian, Zhaoxi Lu, Shuntao Liu, Yinguang Cao.  Virucidal activity and the antiviral mechanism of acidic polysaccharides against Enterovirus 71 infection in vitro. Microbiology and Immunology. 2020, 64 (3); 189-201.

Cagno V, Tintori C, Civra A, Cavalli R, Tiberi M, Botta L, Brai, A., Poli, G., Tapparel, C., Lembo, D., Botta, M. Novel broad spectrum virucidal molecules against enveloped viruses. PLoS ONE. 2018, 13(12): e0208333. https://doi. org/10.1371/journal.pone.0208333

Faral-Tello, P., Mirazo, S., Dutra, C., Perez, A., Geis-Asteggiante, L., Frabasile, S., Koncke, E., Davyt, D., Cavallaro, L., Heinzen, H., Arbiza, J. Cytotoxic, Virucidal, and Antiviral Activity of South American Plant and Algae Extracts. The Scientific World Journa. 2012, 2012, Article ID 174837, 5 pages doi:10.1100/2012/174837

  1. Chiamenti, F. P. Silva, K. Schallemberger, M. Demoliner, C. Rigotto, J. D. Fleck. Cytotoxicity and antiviral activity evaluation of Cymbopogon spp hydroethanolic extracts. Brazilian Journal of Pharmaceutical Sciences. 2019, http://dx.doi.org/10.1590/s2175-97902019000118063

It would be good to also evaluate the simple MRS media for a putative antiviral effect. Maybe it was done while evaluating the IC50, but due to missing data one cannot tell.

Answer: The cytotoxicity and the probability of manifestation of antiviral activity of the MRS medium was investigated and presented in Table 2.

Concerning the evaluation of the virucidal activity shown in Table 3, is there any possibility to use a positive control. It would make the data more reliable than only showing the postmetabolites virucidal activity.

Answer: In this experiment, each value obtained is the result of the decrease in viral titer compared to a viral control incubated under the same conditions and time as the sample. In this experiment, values of change in viral titer Δlg greater than 1.5 were taken as significant, and the greater the value, the greater the virulence. Therefore, we felt that there was no need to present a positive and negative control. However, in the experiment, MRS broth was used as a negative control, and 70% ethyl alcohol as a positive control and are added to the table.

Furthermore, as the virions seem to be quite unstable the authors should provide data to show that after 90 minutes the viruses are still fully infective when not incubated with the substances. It needs to be clarified that the substances to have a virucidal activity at time points that do not change the infectivity per se.

Answer: In the results on page 8 we have mentioned that the last time interval at which the Koi-virions show activity is 90 minutes. At the next investigated time interval 120 min both the samples and virus controls had lost activity, therefore we conclude that outside the cell the Koi-virions can survive only about 90-100 min and the last time interval (120 min is not presented in the results) . When determining the virucidal activity regardless of which virus, it is normal to decrease the titer of the virus over time even without being treated with any substance. This is part of the characteristic of virus particles. Therefore, in this experiment, at each time interval we have a viral control, with which we record the natural decrease of the viral titer. The resulting decrease in viral titer over a given time interval is obtained by subtracting the titer of the treated virus from the titer of the viral control (whose titer also decreases with time).

Why was the pre-treatment of CCB cells performed at 37°C while all other experiments were done at 22°C.?

Answer: I am extremely grateful to the esteemed reviewer for spotting this technical error of ours. All experiments, as well as cell and virus propagation, were performed at 22 °C.

Minor comments:

Page 2: “They comprise a wide range of non-viable biochemical molecules relased by…”

Released instead of relased

Answer: The correction has been introduced in the text.

Page 2: “…antihyperhypertensive…” change to antihypertensive

Answer: The correction has been introduced in the text.

Page 5: “Cytotoxicity assay:  On 72th hour…” change to 72nd

Answer: The correction has been introduced in the text.

Page 11: “Discussion: …glycoproteins of viruses thus again disrupting vual entry.” Change vual in viral.

Answer: The correction has been introduced in the text.

Page 11: “Discussion: The results described by other teams are consistent with our results” The authors need to provide some literature.

Answer: Relevant literary sources are presented in the text.

We hope we've answered your questions and fulfulled your recommendtions.

Round 2

Reviewer 1 Report

Thank you very much for revising our comments.

Reviewer 3 Report

I would like to thank the authors for answering my questions. I do not have further comments.